# Dynamic Visualization of TGF-β/SMAD3 Transcriptional Responses in Single Living Cells

**DOI:** 10.3390/cancers14102508

**Published:** 2022-05-19

**Authors:** Dieuwke L. Marvin, Li You, Laura Bornes, Maarten van Dinther, Niek Peters, Hao Dang, Sarah K. Hakuno, Marten Hornsveld, Onno Kranenburg, Jacco van Rheenen, Jos H. T. Rohling, Miao-Ping Chien, Peter ten Dijke, Laila Ritsma

**Affiliations:** 1Oncode Institute and Department of Cell and Chemical Biology, Leiden University Medical Center, 2333 ZC Leiden, The Netherlands; d.l.marvin@lumc.nl (D.L.M.); m.a.h.van_dinther@lumc.nl (M.v.D.); m.hornsveld@lumc.nl (M.H.); l.m.a.ritsma@lumc.nl (L.R.); 2Oncode Institute and Department of Molecular Genetics, Erasmus University Medical Center, Erasmus MC Cancer Institute, 3015 GD Rotterdam, The Netherlands; l.you@erasmusmc.nl (L.Y.); m.p.chien@erasmusmc.nl (M.-P.C.); 3Oncode Institute and Department of Molecular Pathology, Netherlands Cancer Institute, 1066 CX Amsterdam, The Netherlands; l.bornes@nki.nl (L.B.); j.v.rheenen@nki.nl (J.v.R.); 4UMC Utrecht Cancer Center, University Medical Center Utrecht, Utrecht University, Heidelberglaan 100, 3584 CX Utrecht, The Netherlands; n.a.peters-6@umcutrecht.nl (N.P.); o.kranenburg@umcutrecht.nl (O.K.); 5Department of Gastroenterology and Hepatology, Leiden University Medical Center, 2333 ZC Leiden, The Netherlands; h.dang@lumc.nl (H.D.); s.k.hakuno@lumc.nl (S.K.H.); 6Department of Cell and Chemical Biology, Leiden University Medical Center, 2333 ZC Leiden, The Netherlands; j.h.t.rohling@lumc.nl

**Keywords:** TGF-β signaling, transcriptional reporter, signaling dynamics, signaling heterogeneity

## Abstract

**Simple Summary:**

How a single cytokine can induce a variety of cellular responses in the same cell or in different cells is a longstanding question. Transforming growth factor β (TGF-β) is a prototypical multifunctional cytokine of which biological responses are highly dependent on in a cellular context. TGF-β signals via intracellular SMAD transcription factors, and the duration and intensity of SMAD activation are key determinants for the responses that are elicited by TGF-β. To visualize the TGF-β signaling kinetics, we developed a dynamic TGF-β/SMAD3 transcriptional reporter using a quickly folded and highly unstable green florescent protein. We demonstrate the specificity and sensitivity of this reporter and its wide application to monitor dynamic TGF-β-induced responses in cells cultured on plastic dishes, and in living animals. This tool allows for the analysis of TGF-β signaling at a single living cell level, and allows for the discovery of dynamic TGF-β SMAD- induced transcriptional responses in multi-step biological processes.

**Abstract:**

Transforming growth factor-β (TGF-β) signaling is tightly controlled in duration and intensity during embryonic development and in the adult to maintain tissue homeostasis. To visualize the TGF-β/SMAD3 signaling kinetics, we developed a dynamic TGF-β/SMAD3 transcriptional fluorescent reporter using multimerized SMAD3/4 binding elements driving the expression of a quickly folded and highly unstable GFP protein. We demonstrate the specificity and sensitivity of this reporter and its wide application to monitor dynamic TGF-β/SMAD3 transcriptional responses in both 2D and 3D systems in vitro, as well as in vivo, using live-cell and intravital imaging. Using this reporter in B16F10 cells, we observed single cell heterogeneity in response to TGF-β challenge, which can be categorized into early, late, and non-responders. Because of its broad application potential, this reporter allows for new discoveries into how TGF-β/SMAD3-dependent transcriptional dynamics are affected during multistep and reversible biological processes.

## 1. Introduction

How a single cytokine can induce a wide variety of downstream responses in the same cell or in different cell types is a longstanding question in the signaling field [1,2]. Transforming growth factor β (TGF-β) is a prototypical cytokine in this respect, as its biological responses are highly dependent on cellular context, such as cell type and its differentiation state and/or the presence of other extracellular cues [3,4]. TGF-β is part of a larger family of cytokines that also include activins and bone morphogenetic proteins (BMPs), which signal via selective transmembrane type I and type II serine/threonine kinase receptors and intracellular SMAD transcription factors. Examples of contextual-dependent biological roles include the action of TGF-β family ligands as morphogens during development, in which the SMAD activation duration and intensity are critical determinants. TGF-β/activin can elicit opposing tumor promoting and suppressing effects in cancer [4], which is attributed, in part, to changes in SMAD-interacting (transcription) factors or rerouting through non-canonical signaling pathways [5,6,7,8]. Thus, TGF-β signaling spatio-temporal dynamics are determined by cell intrinsic and extrinsic factors, negative feedback loops, and self-enabling responses; however, how these contribute to different biological outputs is not well understood.

Given the multifunctional properties and wide variety of responses of TGF-β signaling, it is likely that the dynamic control of signaling plays a role [5,9]. Indeed, TGF-β signaling was shown to have a graded early response, and a switch-like sustained response [10]. Others have shown that the response to a TGF-β concentration is transient and adaptive, resulting in an instructive signal for patterning [11]. However, many other questions remain, such as whether dynamic TGF-β signaling responses are heterogeneous, oscillating, and/or coupled to cellular behavior such as cell migration.

Signal encoded states are not merely on-or-off but are highly dynamic [2]. This is likely to be of key importance during multistep biological responses, such as cell invasion, endothelial sprouting, etc. As such, signaling can be encoded by dynamic properties, such as delay, duration, fold-change, or frequency, resulting in extra layers to modulate signaling output [1,2,9]. To detect these dynamic signals, fluorescent-based signaling reporters can be used to visualize and decode these changes in signaling pathways over time [1].

As mentioned above, TGF-β family members signal via intracellular SMAD transcription factors. Whereas activated TGF-β and activin type I receptors mediate the activation of receptor-regulated (R) mother against decapentaplegic homolog 2 (R-SMAD2) and R-SMAD3 by their carboxy (C) terminal phosphorylation on two serine residues, BMP receptors induce the C-terminal phosphorylation of R-SMAD 1, -5, and -8 proteins. Activated R-SMADs form heterodimeric and trimeric complexes with SMAD4 that translocate to the nucleus. The nuclear SMAD complexes bind to specific sequences in promoters of selected target genes, thereby regulating their transcription. The affinity of SMADs for DNA is low, and SMADs require other DNA-binding transcription factors to efficiently bind to target promoters [4]. Target genes for TGF-β/SMAD3 signaling are, among others, *SERPINE1* (Serine proteinase inhibitor, Clade 1, Member 1, which encodes for plasminogen activator inhibitor 1 (PAI-1) protein), *SMAD7*, and connective tissue growth factor (*CTGF)*. Within the promoter region of *SERPINE1*, three 5′-CAGA-3′ boxes were identified as direct binding sites for SMAD3 and 4 [12]. By multimerizing this SMAD Binding Element, a 5′-CAGA-3′ a transcriptional reporter construct was created: 12-times 5′-CAGA-3′ box repeat and a minimal adenovirus major late promoter (MLP) were cloned upfront of a reporter cDNA encoding luciferase [12]. This is a widely used reporter system to interrogate TGF-β/SMAD3 signaling response in cells by preparing cell lysates of in vitro cultures or in vivo by bioluminescence but lacks good temporal or single cell resolution [13,14].

Most signaling studies have been performed using population average methods. The advent of live cell imaging methods and fluorescent tagging of proteins or reporters driving the expression of fluorescent proteins have allowed to monitor signaling dynamics in single cells in real time. TGF-β signaling dynamics have been measured by translocation of fluorescently-tagged (phosphorylated) SMAD2, 3, and 4 from cytoplasm to nucleus, one of the early events of the TGF-β signaling pathway [9,11,15,16,17,18]. These studies and others have provided great insights in direct SMAD shuttling and signaling dynamics. Nuclear translocation of SMAD4, rather than SMAD2, can be correlated to a transcriptional response [15,17]. Transcriptional output can also be determined by other factors than the nuclear translocation of SMAD proteins [19,20].

Fluorescent-based transcriptional reporters allow for single cell visualization over time, and report for transcriptional output of signaling. The latter is more downstream of measuring SMAD2/3 phosphorylation or R-SMAD- or SMAD4 nuclear accumulation, and therefore, more subject to crosstalk with other signaling pathways. For example, SMAD transcriptional activity is determined by the partner transcription factors of which the activation state is dependent on activation by specific stimuli [4]. Luwor et al. coupled a TdTomato fluorophore to a 12x 5′-CAGA-3′ box to monitor effects of TGF-β on breast cancer cell migration in vitro [21]. This TGF-β/SMAD3 fluorescent reporter can give insights in transcription at a single time point and allows for cellular tracking at a single cell level. However, the stability of fluorescent proteins may hinder dynamic visualization over time, and therefore, this (and similar) reporters are not suited for visualizing transcription in a dynamic and temporal manner at a single cell level. Taken together, there is a clear need for a dynamic transcriptional TGF-β reporter to assess TGF-β signaling in vitro, as well as in living animals, at a single cell resolution. Measurement of the TGF-β/SMAD3 transcriptional temporal behavior in living cells will provide new insights into how dynamical features, such as duration, intensity, and frequency of temporal signals, lead to specific cellular responses.

## 2. Materials and Methods

### 2.1. Reagents

TGF-β family ligands, dissolved in 4 mM HCl/0.1% recombinant bovine serum albumin (BSA): TGF-β1 (1 ng/mL unless indicated otherwise, 7666, R&D Systems, Minneapolis, MN, USA), TGF-β2 (1 ng/mL unless indicated otherwise; kind gift from Joachim Nickel, University of Würzburg, Würzburg, Germany), TGF-β3 (1 ng/mL unless indicated otherwise; kind gift from A. Hinck, University of Pittsburgh; this ligand is generally used, unless stated otherwise), Activin A (50 ng/mL, R&D Systems), BMP2 (50 ng/mL; kind gift from Joachim Nickel, University of Würzburg, Würzburg, Germany), BMP6 (50 ng/mL; kind gift from Prof. Slobodan Vukicevic, University of Zagreb, Zagreb, Croatia), BMP7 (50 ng/mL; kind gift from Prof. Slobodan Vukicevic, University of Zagreb, Zagreb, Croatia), BMP9 (50 ng/mL, R&D Systems). Selective small molecule TβRI kinase inhibitor SB505124 (1 mM, DMSO, #3263, Tocris, Bristol, UK) and cycloheximide (50 ng/mL, C1988, Sigma, Darmstadt, Germany) were used. 

### 2.2. Generation of Lentiviral TGF-β/SMAD3 Reporter Constructs

First, the pGLS3-CAGA_12_-LUC plasmid [12,22] was used to generate the pGLS3-CAGA_12_-TdTomato construct by replacing the luciferase gene for the TdTomato gene using *Bam*HI/*Hind*III and *Xba*I. Next, *Cla*I and *Eco*RV/*Sma*I restriction sites were used to clone the CAGA_12_-TdTomato fragment from the pGLS3-CAGA_12_ into a pLV- phosphoglycerate kinase 1 (PGK)-puromycin vector to create the pLV-CAGA_12_-TdTomato reporter, expressing TdTomato under transcriptional control of 12 5′-CAGA-3′ SMAD3 response elements. 

For subcloning, the CAGA_12_-MLP element was amplified through polymerase chain reaction (PCR) using the CAGA_12_-MLP FW and REV (containing *Age*1 restriction site) primers. The purified CAGA_12_-MLP PCR product was digested using *Cla*1 and *Age*I, whereas the pLV vector was digested using *Cla*1 and *Eco*R1. The digested pLV vector was treated with shrimp alkaline phosphatase (SAP) to prevent self-ligation. The isolated pLV vector and CAGA_12_-MLP promoter were used for subsequent cloning.

eGFP was amplified from CMV-d2eGFP-empty (gift from Phil Sharp (Addgene plasmid # 26164; http://n2t.net/addgene:26164 (accessed on 3 September 2021); RRID:Addgene_26164)) [23] through PCR using T7 FW GFP rev (including *EcoR1* restriction site) primers (Appendix A). The PCR product was digested using *Age*1 and *Eco*RI, and was thereafter subcloned with the pLV vector and CAGA_12_-MLP promoter using a 3-way ligation (1:1:1) through incubation overnight at 4 °C with T4 ligase to generate pLV-CAGA_12_-eGFP expressing eGFP under transcriptional control of 12 5′-CAGA-3′ SMAD3 response elements.

To enhance eGFP folding speed and, thus, the appearance of fluorescence signal, we created a third construct with a superfolding GFP (sfGFP) harboring the super folding mutations (S30R, Y39N, N105, Y145F, I171V, A206V) in the eGFP. sfGFP-C1 (gift from Michael Davidson and Geoffrey Waldo (Addgene plasmid # 54579; http://n2t.net/addgene:54579 (accessed on 3 September 2021); RRID:Addgene_54579)) [24] was isolated using *Eco*R1 and *Bsrg*1. Additionally, we engineered a PEST domain at the C-terminus of the construct to promote degradation by the proteasome once TGF-β signaling ceases. CMV-d2eGFP-empty (gift from Phil Sharp (Addgene plasmid # 26164; http://n2t.net/addgene:26164 (accessed on 3 September 2021); RRID:Addgene_26164)) [23] was used to amplify the destabilizing domain through PCR using D2 domain FW and REV primers (Appendix A). The purified product was digested using *Eco*RI and *Bsrg*I. The purified destabilizing domain and sfGFP-C1 products were ligated by incubating at 4 °C with T4 ligase. The resulting dynGFP-C1 was digested with *Age*1 and *Eco*RI to create the dynGFP insertion into the final vector. The pLV vector, CAGA_12_-MLP promoter, and dynGFP were ligated using a 3-way ligation using incubation overnight at 4 °C with T4 ligase. This generated CAGA_12_-dynGFP, expressing superfolder GFP with destabilizing domain under transcriptional control of 12 5′-CAGA-3′ SMAD3 response elements. Plasmid sequences were verified using sanger sequencing using the eGFP FW, eGFP rev, and PGK rev primers (Appendix A).

### 2.3. Cells and Culture Methods

The following cell lines were used: human embryonic kidney cells HEK293T (CRL3216™, ATCC, Manassas, VA, USA), mouse melanoma cells B16F10 (CRL-6475™, ATCC) and YUMM4.1 (CRL-3366™, ATCC), human breast cancer cells MDA-MB-231 (HTB-26™, ATCC), mouse breast cancer D2.OR cells (kind gift from F.R. Miller [25]), normal mammary gland NMuMG derivative NM18 [26], human liver cancer cells (HepG2) (HB-8065™, ATCC), human acute monocytic leukemia cells (THP1) (TIB-202™, ATCC), mouse endothelial cells (MS1) (CRL-2279™, ATCC), pancreatic stellate cells (RLT-PSC) (kind gift from prof M. Löhr and dr. R. Heuchel [27], Stockholm, Sweden), and primary mouse fibroblasts. HEK293T, B16F10, MDA-MB-231, D.2OR, NM18, NIH3T3, HEPG2, and primary mouse fibroblasts were maintained in DMEM containing 10% FBS and 100 U/mL penicillin-streptomycin (15140122; Thermo Fisher Scientific, Waltham, MA, USA). THP1 cells were maintained in RPMI-1640 medium containing 10% FBS and 100 U/mL penicillin-streptomycin (15140122; Thermo Fisher Scientific, Waltham, MA, USA). Mouse melanoma cells (YUMM4.1) and RLT-PSC cells were maintained in Dulbecco’s Modified Eagle Medium DMEM/F12 containing 10% fetal bovine serum (FBS) and 100 U/mL penicillin-streptomycin. MS1 cells were maintained in MEMa containing 10% fetal bovine serum (FBS) and 100 U/mL penicillin-streptomycin. All cell lines were cultured in 37 °C, 5% CO_2_ incubators and routinely tested for the absence of mycoplasma infections. All human cell lines were authenticated using short tandem repeat (STR) profiling.

### 2.4. Generation of CAGA_12_-TdTomato, CAGA_12_-GFP and CAGA_12_-dynGFP Cell Lines

Lentiviruses were generated by HEK293T transfection with packaging constructs and the lentiviral constructs using standard protocols [28]. Cells were exposed to lentivirus containing pLV.CAGA_12_-TdTomato, pLV.CAGA_12_-GFP or pLV.CAGA_12_-dynGFP, and polybrene (8 mg/mL) for 24 h, and were allowed to recover for 1–3 days. Transduced cells were then selected using 1 mg/mL puromycin for >1 week.

B16F10 cells and RLT-PSC cells containing the CAGA_12_-reporter were sorted using the BD FACS Aria III 4L (BD Biosciences, San Jose, CA, USA) at the Flow Cytometry Core Facility (FCF) of Leiden University Medical Center (LUMC) in Leiden, Netherlands, through sorting the top 20% fluorescence-expressing cells after 24 h (B16F10 and RLT-PSC pLV.CAGA_12_-dynGFP) or 48 h (B16F10 pLV.CAGA_12_-TdTomato and B16F10 pLV. CAGA_12_-GFP) of TGF-β3 (1 ng/mL) stimulation. To generate clonal cell lines of B16F10 pLV.CAGA_12_-dynGFP, cells were transduced with lentiviruses containing pLV.CMV-RFP-NLS. Transduced cells were stimulated with TGF-β3 (1 ng/mL) for 24 h prior to FACS sorting of GFP+/RFP+ single cells into a 96-well plate using the BD FACS Aria III 4L (BD Biosciences, San Jose, CA, USA) at the Flow Cytometry Core Facility (FCF) of Leiden University Medical Center (LUMC) in Leiden, Netherlands. Single cells were seeded in one 96-well and expanded, and 3 clonal lines (clone 7, clone 8, clone 9) were selected for further experiments. FACS-sorted cells were used within 1 month of sorting.

### 2.5. Colorectal Cancer Organoid Culture

The colorectal cancer patient-derived organoid (PDO) cultures used in this study were previously established and characterized [29]. The culture conditions used were previously described [29]. Briefly, organoids were cultured in 75% Matrigel^®^ domes (growth factor reduced basement membrane matrix, phenol red-free, Corning^®^ (Corning, NY, USA). The culture medium conditions consist of Advanced DMEM/F12 (Thermo Fisher Scientific, Waltham, MA, USA) with 1% Penicillin/Streptomycin (Thermo Fisher Scientific, Waltham, MA, USA), 10 mM HEPES, 1% Glutamax (Thermo Fisher Scientific, Waltham, MA, USA), 100 ng/mL Noggin conditioned medium, 1X B27, 1.25 mM N-acetyl-L-cysteine, 0.5 mM A83-01 (Tocris), 10 mM SB202190. In general, the organoid medium was refreshed every 2–3 days. After around 1–2 weeks of culture, organoids were harvested and incubated with TrypLE Express (Thermo Fisher Scientific, Waltham, MA, USA) at 37 °C for 1–5 min to obtain a single cell suspension. Residual TrypLE was washed out using Advanced DMEM/F12. Subsequently, the single cells were resuspended in 75% Matrigel/25% Advanced DMEM/F12 domes, and plated in 6-, 12-, or 24- well culture plates.

The PDOs were stably transduced with the pLV.CAGA_12_-dynGFP reporter, and selected with puromycin and subsequent FACS-sorting of 24 h stimulated TGF-β_1_ (5 ng/mL) dynGFP-positive cells. PDOs were dissociated into single cells, and subsequently plated in a 384-well glass bottom plate (4581, Corning). All TGF-β_1_ stimulation experiments were conducted in organoid medium without noggin, A83-01, and N-acetyl-L-cysteine. PDOs were allowed to grow for 7–10 days, after which, the experimental medium (5 ng/mL carrier-free recombinant human TGF-β_1_, R&D 240-B) was added. Organoids were scanned for 72 h with 30 min time intervals using a 40X water objective lens on a Leica SP8X confocal microscope equipped with a culture chamber for adequate conditions (37 °C and 5.0% CO_2_ overflow).

### 2.6. Pancreatic Organoid Co-Culture

Resection-derived pancreatic tumor organoid PDO1 (HUB-08-B2-029B), was obtained from the Hubrecht Organoid Technology Biobank (HUB) [30]. Organoids were cultured in Matrigel^®^ (growth factor reduced basement membrane matrix, phenol red-free, Corning^®^) domes overlaid with advanced DMEM/F12 supplemented with Glutamine/Glutamax (200 nM 100×, Invitrogen, Waltham, MA, USA), 100 UI/mL penicillin, and 100 μg/mL streptomycin (Thermo Fisher Scientific, Waltham, MA, USA), B27 (1x, Invitrogen), recombinant Noggin, 100 ng/mL (Peprotech, London, UK), Nicotinamide 10 mM (Sigma, Darmstadt, Germany), N-Acetyl Cysteine 1.25 mM, (Sigma, Darmstadt, Germany), Primocin 50 μg (Invivogen, San Diego, CA, USA), Gastrin 10 nM (Sigma, Darmstadt, Germany), recombinant R-Spondin 3 200 ng/mL (Peprotech), recombinant FGF10A 100 ng/mL (Peprotech), recombinant EGF (5 ng/mL), A83-01 500 nM (Selleckhem, Munich, Germany), WNT-surrogate–Fc fusion protein 0.5 nM (Utrecht Protein Express, Utrecht, The Netherlands). The pancreatic stellate cell line RLT-PSC was a kind gift from Professor M. Löhr and Doctor R. Heuchel (Stockholm, Sweden) [27]. RLT-PSC cells were transduced with pLentiPGK Hygro DEST H2B-mRuby2 (pLentiPGK Hygro DEST H2B-mRuby2 was a gift from Markus Covert, Addgene plasmid #90236) and selected with 100 mg/mL Hygromycin B (Sigma, Darmstadt, Germany) for one week. Subsequently, RLT-PSC-mRuby2 was transduced with pLV.CAGA_12_-dynGFP and selected with 1 mg/mL Puromycin (Invitrogen). After puromycin selection, cells were stimulated with TGF-β3 (1 ng/mL) for 24 h, and FACs-sorted for the top 20% cells expressing dynGFP.

For co-culture experiments, PDOs and RLT-PSCs were trypsinized to single cell suspensions; PDO and RLT-PSC cells were then mixed together in a 1:10 ratio in RLT-PSC medium containing 1% Matrigel. The mixtures were plated in ultra-low attachment 96-well round-bottom plates (Corning, NY, USA; *n* = 1000 PDO and *n* = 10,000 RLT-PSC cells per well), centrifuged (1200 RPM, 1 min), and incubated overnight (37 °C, 5% CO_2_). The next day, the formed aggregates were collected, and the supernatant was removed. Individual aggregates were taken up in 100% Matrigel and plated in flat-bottom 96-well plates (Greiner, Kremsmünster, Austria) which were pre-coated with 50 mL Matrigel. PDO medium was added after polymerization of the Matrigel. The co-culture experiment was performed without the presence of the TβRI kinase inhibitor A83-01, unless specifically indicated. The different experimental conditions containing either TGF-β1 (5 ng/mL), A83-01 (500 nM, Selleckhem), or control pancreatic organoid medium without A83-01 (endogenous condition) were added and refreshed every 2 days for a period of 5 days. At day 4, A83-01 (500 nM, Selleckhem) was added to the endogenous condition. Photos were taken daily using the Cytation 5 live-cell imaging microscope (Agilent, Santa Clara, CA, USA).

### 2.7. Flow Cytometry Analysis

Cells were treated with TGF-β3 (1 ng/mL), SB505124 (1 mM), cycloheximide (50 mg/mL), or vehicle solvent control for the indicated time points. Samples treated with TGF-β type I receptor kinase inhibitor SB505124 were pre-stimulated with TGF-β 24 h (CAGA_12_-dynGFP) or 48 h (CAGA_12_-TdTomato/CAGA_12_-GFP) prior to SB505124 treatment. To collect cells, cells were trypsinized and resuspended in DMEM containing 10% FBS and pen-strep. Sample were washed and resuspended in PBS containing 0.2% BSA. For cell cycle analysis, cells were treated with Hoechst33342 (20 mg/mL) (#62249, Thermo Fisher) for 45 min at 37 °C prior to trypsinization of cells, and were treated with TGF-β (1 ng/mL) for 2 h prior to trypsinization of cells. Fluorescence was directly measured by BD LSRII (BD Biosciences, Franklin Lakes NJ, USA). Flow cytometry data were analyzed using FlowJo^TM^ (v10, BD Biosciences, Franklin Lakes, NJ, USA).

### 2.8. rt-qPCR

B16F10 CAGA_12_-TdTomato/CAGA_12_-GFP/CAGA_12_-dynGFP cells were treated with TGF-β3 (1 ng/mL) and/or SB505124 (1 mM). Samples treated with SB505124 were pre-stimulated with TGF-β 24 h (CAGA_12_-dynGFP) or 48 h (CAGA_12_-TdTomato/CAGA_12_-GFP) prior to SB505124 treatment. Total RNA extraction was performed using the NucleoSpin RNA II kit (Macherey-Nagel, Dűren, Germany). An amount of 1 mg of RNA was used for cDNA synthesis using the Revert Aid First Strand cDNA synthesis kit (Thermo Fisher). Quantitative PCR was performed using SYBBR GoTaq qPCR master mix (Promega, Madison, WI, USA) and 0.5 mM of primers targeting (dyn)GFP, SMAD7, *Serpine1*, *Ctgf*, *Gapdh*, and *Hprt* (Appendix A). RT-qPCR was performed on the CFX Connect Real-Time PCR detection system (Bio-Rad, Hercules, CA, USA). Measurements were performed in technical duplicate and independent biological triplicate; target gene expression was normalized for *Gapdh* and *Hprt* expression.

### 2.9. Western Blotting

Protein lysates were harvested in Laemmli buffer (0.12 M Tris-HCl pH 6.8, 4% SDS, 20% glycerol, 35 mM β-mercaptoethanol, and bromophenol blue) and boiled for 5 min. Western blotting was performed using standard procedures. Membranes were blocked in 5% non-fat dry milk for 1 h at room temperature and incubated with primary antibody in 2.5% non-fat dry milk overnight at 4 °C. The primary antibodies used were GFP (sc-8334, Santa Cruz), PAI-1 (ab222754, Abcam, Cambridge, UK), phospho-SMAD1 and 2 [31], GAPDH, and Vinculin (V9131, Sigma, Darmstadt, Germany). As the C-termini of SMAD3, and BMP R-SMAD1 and -5 are identical, p-SMAD2 was used to assess TβRI activity. Experiments were performed in independent biological triplicate; representative images are shown.

### 2.10. CAGA_12_-Luciferase Assay

B16F10 cells were seeded in a 24-well plate and transfected with plasmids containing the CAGA_12_-luc transcriptional reporter and SV40 promoter-controlled β-Gal cDNA expression vector expressing β-galactosidase using *polyethylenimine* (PEI). Cells were washed and stimulated with TGF-β3 (1 ng/mL) and/or SB505124 (1 mM) overnight. Cells were lysed, and luciferase and β-galactosidase signal were measured with a PerkinElmer plate reader. Luciferase signals were corrected for signals in untransfected control samples and normalized for b-galactosidase expression. Experiments were performed by integrating results of three independent wells.

### 2.11. Live Cell Imaging Using Incucyte^®^

CAGA12-GFP/CAGA12-dynGFP reporter B16F10 cells were plated in a 96-well plate (#3595, Corning) and were stimulated with TGF-β3 (1 ng/mL or otherwise indicated), SB505124 (1 mM), cycloheximide (50 mg/mL), or vehicle solvent control for the indicated time points. Measurements were taken every 2 h using the Incucyte^®^ S3 live-cell analysis system using brightfield and a GFP laser with 300 msec exposure. Fluorescence intensity and confluency were measured using Incucyte^®^ S3 Software. Experiments were performed by integrating results of three independent wells. Signal over background (s/b) scores were calculated by dividing the average fluorescence intensity upon TGF-β stimulation by unstimulated controls at the indicated timepoints. Z′ scores were calculated using the following formula:$$Z^{\prime }=\frac{3\times \left({\mathrm{STDev}}_{\mathrm{pos}}+{\mathrm{STDev}}_{\mathrm{neg}}\right)}{\left|\mathrm{AVGpos}\;-\;\mathrm{AVGneg}\right|}$$

### 2.12. Confocal Microscopy

B16F10 CAGA_12_-TdTomato/CAGA_12_-GFP/CAGA_12_-dynGFP were plated in a chambered coverslip (80826 Ibidi, Gräfelfing Germany). Cells were treated with TGF-β3 (1 ng/mL) and/or SB505124 (1 mM) for 24 h (CAGA_12_-dynGFP) or 48 h (CAGA_12_-TdTomato/CAGA_12_-GFP). Live cells were stained with Hoechst 33342 (10 mg/mL) (#62249, Thermo Fisher) for 15 min, prior to imaging. Confocal images were taken on a Leica TCS SP5 confocal laser scanning microscope.

### 2.13. Intravital Imaging Experiments

#### Experimental Model and Subject Details

All mice were housed under a 12 h light/dark cycle and under specific pathogen-free laboratory conditions, receiving food and water ad libitum. All experiments were approved and performed according to the guidelines of the Animal Welfare Committees of the Netherlands Cancer Institute, protocol code 9.1.957, The Netherlands. *MMTV*-PyMT, *R26*-mTmG, and E-cad-mCFP transgenic mice were crossed to generate donor organoids, i.e., MMTV-*PyMT* and *R26*-mTmG mice [32,33] (mice were purchased from Jackson Laboratory, Sacramento, CA, USA); Ecad-mCFP mice [34] were a gift from Hans Clevers. The process of isolation and culturing of the donor mouse organoids are described in [35]. As acceptors for orthotopic transplantation, 8-to-16 weeks old female Friend Virus B NIH Jackson mice (referred in the text to as FVB) were used. For transplantation, 250,000 single cells were plated 3 days prior to transplantation. At the day of transplantation, BME was dissolved by mechanical disruption, and cells were dissolved in 100 μL Basal Membrane Extract (BME) type 2 (RGF BME type 2 PathClear):PBS, and injected into the fat pad of the 4th mammary gland of acceptor mice.

### 2.14. PYMT Organoid Generation and Viral Transductions

*MMTV*-PyMT, *R26*-mTmG, and E-cad-mCFP transgenic mice spontaneously developed mammary tumors at the age of 8–14 weeks. Upon tumor formation, mice were sacrificed, and mammary tumor organoids were isolated from one donor. Mammary tumors were minced and enzymatically digested: gently shaken for 30 min at 37 °C in digestion mix (0.2% trypsin (from bovine pancreas, Sigma, Darmstadt, Germany) and 0.2% collagenase A (Roche, Basel, Swtizerland). The digested tumors were spun down, and cell fragments were embedded in Basal Membrane Extract (BME) type 2 (RGF BME type 2 PathClear). The mammary tumor organoid medium contained DMEM/F12 Glutamax (Thermo Fisher Scientific, Waltham, MA, USA, 2% B27 (Invitrogen), and 10 ng/mL FGF.

For lentiviral production, HEK293T cells (ATCC^®^ CRL-1573™), were plated at 7 × 10^6^ cells per 15 cm dish. After 24 h, cells were transfected with 32 mg of the pLV. CAGA_12_-dynGFP lentiviral plasmid, and the lentiviral packaging and envelope expression plasmids (pMDL, REV, VSV-G) were diluted in 1 mL of Optimem (Gibco) and 150 mL PEI. After mixing and 5 min incubation at RT, the mixture was added dropwise to the dish. Two days after transfection, the supernatant of transfected cells was harvested, filtered through 0.45 mm filters, and concentrated using Amnico Ultra centrifugal filter units (Merck, Darmstadt, Germany). For lentiviral organoid transduction, organoids were dissociated into single cells by 10 min digestion at 37 °C in TrypLE (Thermo Fisher Scientific, Waltham, MA, USA), followed by mechanical disruption. Single cells were resuspended in concentrated virus supplemented with Y-27632 (1/1000, #M1817, Abmole Bioscience, Houston, TX, USA) and polybrene, and spin-incubated at RT for 60 min at 600 rpm in a 48-well plate. After incubation for 6 h at 37 °C, the cell suspension was collected, and single cells were embedded in BME. The mammary tumor organoid medium was supplemented with Y-27632 (1/1000, #M1817, Abmole) until organoids recovered.

After recovery and expansion, organoids were treated for 48 h with 5 ng/mL TGF-β1. Organoids were collected, washed, and dissociated into single cells by enzymatic digestion for 10 min at 37 °C in TrypLE and mechanical disruption. Cells were resuspended in FACS buffer (PBS + 5 mM EDTA, 2% FBS), and the mTdTom^+^/GFP^+^ cell population was sorted on a FACS ARIA Fusion (BD biosciences) to >90% purity.

### 2.15. Single Cell Live Cell Imaging

Before conducting experiments, 100K cells B16F10 CAGA_12_-dynGFP containing a pLV.CMV-RFP-NLS were seeded on fibronectin (0.1 mg/mL)-coated 14 mm-glass bottom dishes in DMEM medium without phenol red. Experiments were performed 16–24 h after plating on the glass-bottom dishes. The cells were simultaneously imaged using 460 nm (dynGFP; 10 mW/cm^2^) and 532 nm (RFP-NLS10; mW/cm^2^) excitation for 1 day at 1 frame per 3 min with the 1x objective (numerical aperture = 0.5) in the Ultrawide field-of-view optical microscope [36]. The field of view size (~2.5 × 3.5 mm) was chosen for imaging containing a fraction of the cells. For the Clone 7 data, 2575 cells were detected by the end of day 1; for the Clone 9 data, 10748 cells were detected by the end of day 1. We have successfully tracked 83.1% (=2139/2575) of the Clone 7 population, and 96.9% (=10,414/10,748) of the Clone 9 population.

The movement of CAGA_12_-dynGFP B16F10 cells expressing red fluorescent protein (RFP) containing a nuclear localization signal (NLS) was tracked by RFP signal and analyzed using the mTGMM (modified Tracking using Gaussian Mixture Model) cell-tracking algorithm [36]. In brief, the intensity profile of individual nuclei/cell was modelled as a 2D Gaussian distribution. Nuclei tracking was performed by forwarding every Gaussian from time point *t* to *t* + 1 using Bayesian inference, with a priori knowledge that the position, shape, and overall intensity of nuclei could not change dramatically between two consecutive time points. After nuclei/cell detection and tracking, nuclei masks and positions over time were recorded. The RFP signal of individual cells was then extracted from the GFP channel over time using the nuclei/cell masks.

After segmentation and tracking of the cells, the GFP signal of individual cells from the GFP channel over time was extracted. A heatmap was generated to visualize the dynamical process over all cells of this dataset. The dataset could be further classified into groups via Euclidean distance-based hierarchical clustering analysis. After the analysis, two distinct groups were clearly observed: the first group had the GFP intensity trace going up at around 300 min, and the second group had the GFP intensity trace going up at around 700 min. We, therefore, called these two groups the early and late responders, respectively.

### 2.16. Statistical Analysis

Experiments were performed in independent biological triplicates, unless otherwise indicated. Error bars represent ± standard error of the mean (SEM), unless otherwise indicated. Statistics were calculated using GraphPad Prism 7 software.

## 3. Results

### 3.1. Generation of a Dynamic TGF-β/SMAD3 Transcriptional Reporter, CAGA_12_-dynGFP

With the aim to generate a TGF-β/SMAD3 transcriptional reporter that is capable of monitoring signaling dynamics at the transcriptional level in single living cells, we created the lentiviral pLV-CAGA_12_-sfGFP-PEST-PGK-Puro vector, also referred to as the dynamic GFP (CAGA_12_-dynGFP) reporter. To enhance the appearance and clearance of the fluorescence signal compared to the enhanced (e)GFP reporter (pLV-CAGA_12_-EGFP-PGK-Puro), we used an eGFP with so-called superfolding mutations (S30R, Y39N, N105, Y145F, I171V, A206V) [24], and fused a PEST domain (a peptide sequence that is rich in proline (P), glutamic acid (E), serine (S), and threonine (T) at the eGFP C-terminus to promote degradation by the proteasome once TGF-β-induced transcription ceases (Figure 1A) [23,37]. To compare our fast-folding, fast-degrading CAGA_12_-dynGFP with the previously used fluorescent CAGA_12_-GFP and CAGA_12_-TdTomato reporter, we additionally created a pLV-CAGA_12_-GFP-PGK-Puro vector and a pLV-CAGA_12_-TdTomato-PGK-Puro vector.

To assess the kinetic responsiveness of the transcriptional reporters to TGF-β, the constructs were transduced by lentiviral infection into the murine melanoma cell line, B16F10. The B16F10 cells were found, based on TGF-β-induced CAGA_12_-luciferase transcriptional reporter assay and SMAD2 phosphorylation, to be highly responsive to TGF-β stimulation, while having no background TGF-β signaling in unstimulated conditions (Appendix A). As expected, the cells stably expressing the CAGA_12_-TdTomato, CAGA_12_-GFP, and CAGA_12_-dynGFP showed fluorescence reporter activity after the addition of TGF-β, and not when the cells were co-treated with a selective TβRI kinase inhibitor SB505124 and TGF-β (Figure 1B). The 5′-CAGA-3′ DNA boxes were previously reported to be specific for SMAD3 and SMAD4 binding, and thus, respond to TGF-β isoforms and activin, but are unresponsive to BMP cellular stimulation. Consistent with these results, we observed that the CAGA_12_-dynGFP reporter cells responded to TGF-β1,-β2, -β3, and Activin A (albeit weakly), but not to BMPs (Appendix A). The treatment of B16F10 CAGA_12_-dynGFP cells with TGF-β1 or TGF-β3 resulted in a similar dynGFP fluorescence increase (Appendix A). The differential level of fluorescence signal induced by TGF-β isoforms and activin correlated with the differential ability of these ligands to induce SMAD2 phosphorylation (Appendix A).

As the first experiment to assess TGF-β-induced transcriptional responses with CAGA_12_-dynGFP, we compared this reporter side-by-side to the widely used CAGA_12_-luciferase reporter in B16F10 cells. Indeed, the CAGA_12_-dynGFP reporter showed similar results using an end-point assay on lysed cells by plate reader (Appendix A). Additionally, protein levels of dynGFP and target protein PAI-1 followed similar trends upon TGF-β and SB505124 treatment (Appendix A).

Combined, these results suggest that the CAGA_12_-dynGFP reporter is a specific TGF-β/SMAD3 transcriptional reporter which is as efficient in visualizing the TGF-β/SMAD transcriptional response as the widely used CAGA_12_-luciferase reporter.

### 3.2. CAGA_12_-dynGFP Reporter Shows Improved Dynamics

Next, we assessed the dynamic characteristics of the CAGA_12_-dynGFP reporter, and compared its activity with the previously used CAGA_12_-TdTomato and CAGA_12_-GFP reporters [18,21,38] in B16F10 cells. Importantly, CAGA_12_-dynGFP demonstrated a quicker increase and peak intensity of fluorescent signal upon TGF-β treatment than the TdTomato and GFP reporter (Figure 1C). After 4 h of TGF-β treatment, 92% of CAGA_12_-dynGFP cells were fluorescent, which is a significantly faster response when compared to CAGA_12_-GFP (25%) and CAGA_12_-TdTomato (29%) (*p* < 0.0001) (Figure 1D). Similarly, the CAGA_12_-dynGFP reporter showed a significantly faster decrease in fluorescently positive cells upon treatment with TβRI kinase inhibitor SB505124 when compared to CAGA_12_-GFP and CAGA_12_-TdTomato (*p* < 0.0001) (Figure 1D). When only 23% of CAGA_12_-dynGFP reporter cells still showed fluorescent signal after 24 h of SB505124 treatment, 90% and 86% of cells containing the CAGA_12_-TdTomato and CAGA_12_-GFP reporter, respectively, were still fluorescent (Figure 1D). Comparing the area under the curve (AUC) of the time course of the different reporters in Figure 1D, the AUC for CAGA_12_-dynGFP was significantly increased in TGF-β treated conditions, and significantly decreased in SB505124 treated conditions, compared to CAGA_12_-TdTomato and CAGA_12_-GFP (all comparisons *p* < 0.0001). Combined, these results shows that the CAGA_12_-dynGFP reporter is superior to both the CAGA_12_-GFP and CAGA_12_-TdTomato in reporting TGF-β/SMAD3 signaling in a dynamic manner.

The decrease of dynGFP signal in pre-stimulated cells after 24 h of SB505124 treatment was more pronounced as compared to GFP, although it was slower than initially expected. When cells are treated with SB505124, TGF-β signaling is blocked at the TGF-β type I receptor level. If reporter mRNA is present, this still can result in reporter protein synthesis. To assess if this is indeed the case, we determined the half-life of the dynGFP fluorescent reporter protein by inhibiting protein synthesis using cycloheximide (CHX), and compared it to a TGF-β washout, or SB505124 treatment (Figure 1E,F). Cells were pre-stimulated with TGF-β for 24 h so that fluorescence would reach maximal levels, after which, the cells were treated with SB505124 or CHX for different time points. Interestingly, we observed that after 4 h of CHX treatment, the fluorescent dynGFP signal decreased to 48%, compared to 90% with SB505124 treatment, as measured by western blotting (Figure 1E) and flow cytometry (Figure 1F). This indicates that upon blocking TGF-β signaling with SB505124, there is a window in which dynGFP protein synthesis still occurs.

As CHX treatment allows for measuring the half-life of a fluorescent protein, we next compared the breakdown and half-life of the three different reporters. Again, cells were pre-stimulated with TGF-β for 24 h (CAGA_12_-dynGFP) or 48 h (CAGA_12_-TdTomato/GFP) so that fluorescence would reach maximal levels, after which, they were treated with CHX. For the CAGA_12_-TdTomato and CAGA_12_-GFP reporters, fluorescence signals reduced minimally (~20% drop in fluorescence after 8 h of CHX, so protein half-lives were estimated to be >24 h) (Figure 1G). In contrast, dynGFP showed an ~80% drop in fluorescence after 8 h of CHX, and a half-life of ~4 h (Figure 1E). Combined, these data again highlight that the CAGA_12_-dynGFP reporter is superior to both the CAGA_12_-GFP and CAGA_12_-TdTomato in reporting TGF-β/SMAD3 signaling in a dynamic manner.

### 3.3. Use of TGF-β Reporter for Live Cell Imaging

Our CAGA_12_-dynGFP reporter has the benefits of lentiviral-based vectors and allows for the efficient delivery in a broad array of dividing and non-dividing cells. We validated the use of CAGA_12_-dynGFP reporter as a sensitive and dynamic transcriptional reporter system in various cell lines (attachment and suspension cells, cancer cell lines, non-cancer cell lines, and primary cells) (Figure 2A). All cells displayed kinetic activation responses of the reporter, measured on the Incucyte^®^ live cell imaging system, upon treatment with TGF-β for 24 h (Figure 2B). We calculated the signal over background (s/b) and Z factor (Z′ score, Appendix A), both important parameters of assay quality when screening compounds for drug discovery. In general, an assay is deemed of sufficient quality for compound testing with an s/b > 2, and a Z′ score > 0.5 [39]. Indeed, we exceeded those numbers in all but two of the cell lines tested (Appendix A). This suggests that the reporter is a useful tool for compound screening using (high content) live cell imaging (screening) platforms such as the Incucyte^®^.

In Appendix A, we followed the effect of different concentrations of TGF-β on dynGFP signal over the course of 3 days in HepG2, NIH3T3, and B16F10 cells. Higher TGF-β concentrations result in later and longer peaks of dynGFP signal for all cell lines, and a decrease of dynGFP signal at later timepoints. Interestingly, the treatment of HepG2 with 10 ng/mL TGF-β results in a second peak, which is easily identified with live cell imaging. These results show that the dynGFP reporter can be used for drug discovery, as well as dynamic monitoring of TGF-β signaling in both attachment and suspension cells, as well as cancer cell lines, non-cancer cell lines, and primary cells.

### 3.4. TGF-β Reporter Shows Heterogenic and Dynamic Responses to TGF-β

Heterogeneity in intensity, duration, or other characteristics of the TGF-β/SMAD3 transcriptional temporal signal could determine cellular response at the single cell level [36]. Therefore, we assessed if cell variation in the temporal behavior of the TGF-β /SMAD3 transcriptional response could be observed at a single cell level using our dynGFP reporter. We combined our dynamic TGF-β transcriptional reporter with a state-of-the-art single cell microscopy technique, where we can automatically segment and track the dynGFP signal in individual cells in real time, in thousands of cells simultaneously. When examining the dynGFP fluorescent intensity within a clonal reporter cell line, we noticed a heterogeneity in signal intensity between cells at a single time point. This is most likely related to slight clonal drift, resulting in differences in gene silencing or expression, and not related to TGF-β response. To assess this, we created a clonal B16F10 SFFV-dynGFP control cell line which constitutively expresses the dynGFP fluorophore. Upon measuring dynGFP signal in the B16F10 CAGA_12_-dynGFP and SFFV-dynGFP clonal lines, we could indeed confirm similar heterogeneity in signal intensity between the lines, concluding that this was the result of technical variation (Appendix A).

We then examined temporal responses in single cells upon TGF-β stimulation. Using clonal B16F10 cell lines containing the CAGA_12_-dynGFP reporter, the dynGFP signal in individual cells was tracked upon treatment of cells with TGF-β over 24 h (1440 min). We observed variation in the temporal pattern of response over time. When the fluorescent signal was normalized between 0 (blue) and 1 (red) for each cell, presented in a heatmap (Figure 3A), we observed different signaling patterns in TGF-β/SMAD3 transcriptional response. Upon clustering of the different temporal responses of the single cells, different response groups within the clonal cell line could be observed (Appendix A). In these groups, three main clusters could be identified: early responders (Figure 3B,E), late responders (Figure 3C,E), and non-responders (Figure 3D). Early responders peaked earlier in their TGF-β reporter response compared to late responders (Figure 3E,F). Similar clustering for early, late, and non-responders were observed for the other investigated B16F10 CAGA_12_-dynGFP clonal line (Appendix A).

Thus, using our CAGA_12_-dynGFP reporter, we can visualize dynamics in TGF-β induced transcription in single cells over time. Using this approach, we can cluster different signaling responses which could have consequences for the biological outcome of the signaling.

### 3.5. Application of TGF-β Reporter in 3D Cell Culture Systems

Organoids are three-dimensional (3D) in vitro cell cultures that resemble some of the key features of an organ. They are important tools that more closely mimic in vivo biology than 2D or 3D spheroid-type co-cultures, and, especially, colorectal cancer organoids are widely used [40]. About 40% of colorectal cancer (CRC) patients harbor mutations in the TGF-β signaling pathway, often rendering SMAD3/4 canonical transcription unresponsive [41,42]. However, 60% of CRC patients have an intact TGF-β/SMAD3 canonical pathway. We stably introduced the CAGA_12_-dynGFP reporter in colorectal cancer organoids that did not harbor mutations in TGF-β signaling pathway components by lentiviral transduction and assessed the SMAD3-driven transcriptional response after TGF-β stimulation. No or limited endogenous TGF-β signaling occurred in these organoids, as no fluorescence was observed at baseline (Figure 4A,B), but upon addition of exogenous TGF-β, CAGA_12_-dynGFP accumulation was observed in the entire organoid (Figure 4A,B). A decrease of the signal could be observed over time.

This indicates that the dynGFP reporter can be used to assess TGF-β transcriptional signaling kinetics in a colorectal cancer organoid model.

### 3.6. Visualizing Endogenous TGF-β Signaling In Vitro and In Vivo

The tumor microenvironment, including fibroblasts, plays an important role in supporting tumor growth [43]. Latent TGF-β in the tumor microenvironment can be locally activated by the interplay between cancer cells and fibroblasts, creating a local niche containing active TGF-β signaling [4,43]. Using the CAGA_12_-dynGFP fluorescent reporter, we aimed to visualize endogenous TGF-β signaling within fibroblast-containing tumor microenvironments.

First, we assessed if the CAGA_12_-dynGFP reporter is able to report endogenous TGF-β activity in a pancreatic cancer organoid model co-cultured with pancreatic stellate cells (RLT-PSCs). Pancreatic cancer organoids that are wildtype in TGF-β pathway components were used, and RLT-PSCs expressing H2B-mRuby were transduced with the dynGFP reporter and selected using FACS sort. On day 0 of the experiment, organoid co-cultures were treated with TGF-β3, A83-01 (TGF-β receptor I inhibitor), or endogenous conditions. After stimulation of the co-culture with TGF-β for 3 and 5 days, activation of the dynGFP reporter was seen in some, but not all fibroblasts (Figure 4C). Upon TGF-β stimulation, dynGFP signal could be seen in RLT-PSCs on the rim, as well as within the organoid core (Figure 4C). With the addition of A83-01, no dynGFP signal was observed. Interestingly, under endogenous conditions, dynGFP signal could be observed in RLT-PSCs in the core of the organoids (Figure 4C). When blocking TGF-β signaling with A83-01 after 4 days of endogenous conditions, the endogenous dynGFP signal could be reduced (Figure 4C), indicating that the dynGFP signal is indeed TGF-β dependent. This suggests that the dynGFP reporter is capable of picking up endogenous levels of TGF-β.

To test whether the CAGA_12_-dynGFP reporter can be used to visualize endogenous TGF-β signaling in vivo as well, the reporter was transduced into PYMT organoids derived from MMTV-*PyMT* and *R26*-mTmG mice, which were subsequently FACS-sorted for dynGFP expression (Appendix A) [32,33]. In vitro, endogenous TGF-β signaling was observed in only few cells within the PyMT organoids (Appendix A). After TGF-β treatment, an increase of dynGFP expression was observed, resulting in a heterogenous intensity pattern of dynGFP. Upon transplantation of these organoids into the mammary fat pad of FVB mice to create primary mammary tumors, intravital imaging was performed through surgical exposure of the tumors. Membranous TdTomato (mTdTomato) is shown in magenta, and Second Harmonic Generation (SHG, in grey) represents the type-I collagen network to provide structural information. In vivo, endogenous CAGA_12_-dynGFP signal was observed in the PyMT primary tumors (Figure 4D). A heterogenous pattern in dynGFP signal was observed throughout the tumors, similar to the heterogeneity observed upon stimulation with TGF-β in vitro (Figure 4D, zoom, Appendix A). In summary, the lentiviral CAGA_12_-dynGFP construct can be widely applied in different 2D and 3D culture systems, as well as in biologically relevant contexts in vivo.

## 4. Discussion

TGF-β is a highly context dependent cytokine that elicits its multifunctional effects via intracellular SMAD transcriptional effectors, of which the activity is determined by the integration of diverse extracellular and internal cues. Visualization of the dynamic aspects of TGF-β/SMAD-induced transcriptional responses will allow for refining and provide novel insights into the role of temporal behavior of SMAD signaling in the physio-pathological processes and consequences of pharmacological/(epi)genetic manipulation. However, the folding and degradation time of fluorescent proteins limits the ability of fluorescent transcriptional reporters to reflect dynamics in signaling. By using a quickly folded and highly unstable fluorescent GFP, we were able to create a TGF-β/SMAD3 reporter that allows the monitoring of SMAD3-driven transcriptional temporal responses in single living cells. Using this reporter in a clonal B16F10 melanoma cell line, we observed heterogeneity in TGF-β/SMAD3-induced transcriptional response, distinguishing slow, fast, and non-responders. We further highlight the use of this reporter in different 2D cell cultures and 3D organoid (co-) cultures, as well is in a biologically relevant context in vivo using intravital imaging. These observations will promote studies in which the activity/expression or localization of signaling molecules requires (single cell) time-lapse imaging of thousands of living cells, and studies on (endogenous) TGF-β signaling dynamics in vitro and in vivo.

To study TGF-β/SMAD-induced transcription, luciferase-based reporters coupled with multimerized SMAD3/4 binding elements have been used extensively in the literature, both in vitro and in vivo [11,12,15,44,45,46,47]. Zhu and colleagues used the CAGA_12_-luciferase reporter in breast cancer cells to study TGF-β/SMAD3 activity by measuring luminescence during tumorigenesis in vivo, whereas others created a transgenic CAGA_12_-luciferase mouse, studying TGF-β/SMAD3 response to injury [45,48]. To achieve a more dynamic visualization with the CAGA_12_-luciferase reporter, Sorre and colleagues coupled a nano luciferase (NLuc) as reporter to the CAGA_12_ box and used microfluidic technology to apply an unstable Nluc substrate prior to image signal readout [11]. This setup, however, requires specialized microfluidics equipment and operator expertise, and is therefore not easily adaptable by other laboratories. Moreover, luciferase-based assays in vivo do not allow for a single-cell resolution.

Fluorescent transcriptional reporters allow for single-cell resolution. However, since the protein half-life of fluorophores can be long (e.g., >24 h for GFP), signaling dynamics cannot be appropriately studied with the current reporters. The half-life of our new dynGFP reporter is significantly reduced (~4 h), and is therefore, in combination with enhanced protein folding, better suited to study signaling dynamics. The sensitivity of our reporter will, however, be dependent on the sensitivity of the experimental readout, the sensitivity of the cell line, and the level of integration of the lentiviral reporter. The current improvements were made on the protein level. Additionally, to enhance instability of the dynGFP reporter on an mRNA level, an unstable polyA tail could be added [49]. However, this is not suited for lentiviral vectors, as an additional polyA tail behind the gene of interest in lentiviral vectors reduces viral titers.

The CAGA_12_-dynGFP reporter can be used in live cell imaging systems, such as the Incucyte^®^, to assess the dynamic response to TGF-β in a bulk population of cells. This is a very suitable set-up for examining, e.g., the effect of drug responses on TGF-β activity. However, this system is less qualified to observe different responses to TGF-β between single cells, due to its limited resolution and capability to track single cells over time. Using microscope systems specialized for single cell imaging, CAGA_12_-dynGFP increases can be monitored in single cells to study TGF-β responses over time.

Some limitations of the reporter should also be discussed. We used clonal B16F10 CAGA_12_-dynGFP cells to investigate TGF-β responses throughout the cell cycle. We observed a higher CAGA_12_-dynGFP signal in later phases of the cell cycle; however, using a clonal control cell line constitutively expressing dynGFP, we observed a very similar pattern. We therefore encourage the use of a constitutive control reporter in addition to the dynGFP for cell-cycle related studies.

The dynamic TGF-β/SMAD3 reporter is highly suited for investigating dynamic responses to TGF-β in different settings. However, as heterogeneity in dynGFP intensity within a (clonal) CAGA_12_-dynGFP cell line can exist for technical reasons, we encourage the use of the reporter for studying TGF-β responses over time, in contrast to comparing reporter signal at one time point. Lastly, we noted a difference in fluorescence reporter activity 48 h after a single TGF-β treatment (when comparing Figure 1C and Appendix A). We hypothesize that these differences are the result of differences in confluency and well size, which resulted in a different TGF-β turnover. Proper controls should be included when interpreting these results.

Visualizing endogenous dynamic TGF-β/SMAD3-induced responses without the addition of exogenous TGF-β in biologically relevant environments in vitro or in vivo may provide novel insights. Based on our in vitro results, the reporter can pick up (exogenously stimulated) TGF-β levels as low as 0.04 ng/mL in various cell lines. This is sensitive enough to pick up endogenous TGF-β levels in tissues, as we observed dynGFP+ cells in organoid co-cultures of fibroblasts and pancreatic cancer cells, and in vivo in mouse mammary tumors. This reporter can visualize both paracrine and autocrine signaling, and it will be interesting to distinguish between the two using additional controls. In mouse mammary tumors, we could observe a heterogenous pattern of TGF-β signaling within the tumor. These heterogeneous patterns were also observed in vitro upon TGF-β stimulation, showing that the in vivo (endogenous) TGF-β activation mimics TGF-β stimulation in vitro. As the cells were not sorted to a single cell clone, this heterogeneity could be due to differences in incorporation of the reporter (Figure 3). Alternatively, the heterogeneity in intensity is the result of heterogeneity in response, similar to what Zhu and colleagues observed when treating breast cancer cells with TGF-β. These researchers used the CAGA_12_-TdTomato reporter in MDA-MB-231 breast cancer cells and observed a population showing no TdTomato signal upon TGF-β treatment, as well as populations showing low, medium, and high TdTomato signal. It would be interesting to use the dynamic CAGA_12_-dynGFP reporter in this case to follow these dynamics over time [21]. Giampieri et al. used a CAGA_12_-CFP reporter and SMAD2 nuclear translocation to monitor TGF-β signaling in primary breast cancer and lymph node metastasis [18]. They also reported a heterogeneity in CAGA_12_-CFP signal in primary MTLn3E tumors, similar to the heterogeneity we observed in PYMT tumors. Additionally, Giampieri reported that, upon metastasizing to the lymph node, breast cancer cells lost their motility. Motile cells in the primary tumor were mostly CAGA_12_-CFP-positive. Interestingly, once metastasized to the lymph nodes, they observed mostly non-motile, CAGA_12_-CFP negative cells in late lymph node metastasis, suggesting that the TGF-β/SMAD3 transcriptional response during lymph node metastasis is transient [18]. Taking the results together, it would be of interest to examine the contribution of TGF-β/SMAD3 signaling dynamics during lymph node metastasis of breast cancer cells containing the CAGA_12_-dynGFP reporter by performing long-term intravital imaging.

## 5. Conclusions

In conclusion, the new TGF-β reporter presented in this study is beneficial over existing TGF-β transcriptional reporters because of the enhanced folding and degradation of the dynGFP protein. This allows for new (drug) discoveries in TGF-β signaling dynamics. The lentiviral backbone and the simplicity of the system make it easily adaptable to different cell systems, including single cell imaging, 3D organoid culture, and intravital imaging. The easy and quantitative read-out of CAGA_12_-dynGFP signal in stably transfected cells using live-imaging high-content experiments can be readily performed. Its robust signal allows for high quality assays to be used in drug discovery. Because of the many roles of TGF-β in different processes and cell types, this reporter can be of value in many fields of biological research.

## Figures and Tables

**Figure 1 cancers-14-02508-f001:**
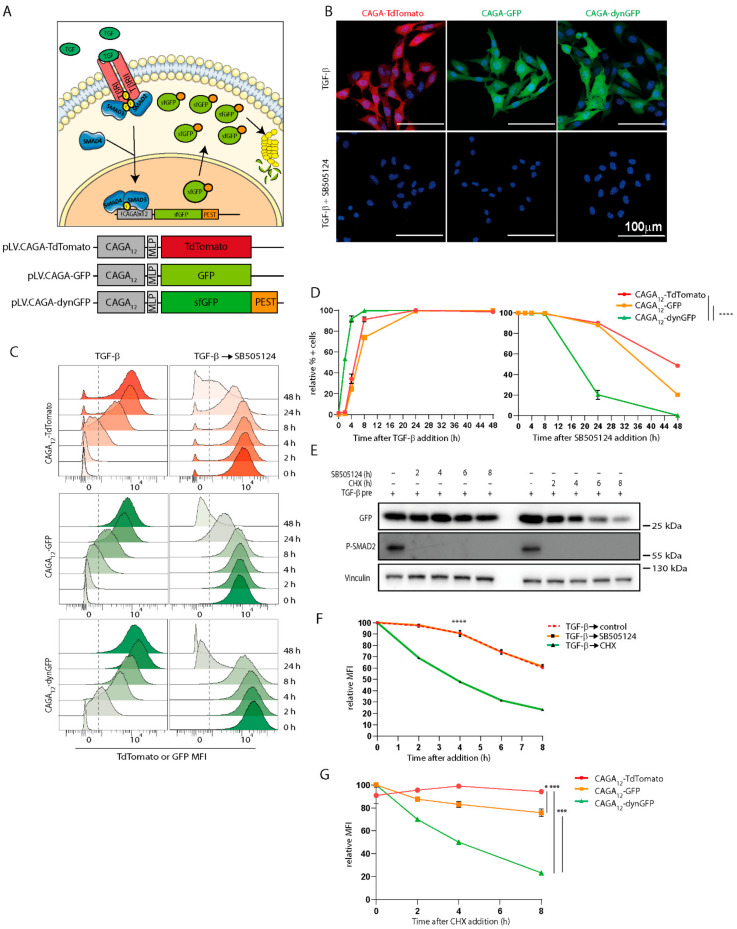
Generation and validation of the dynamic TGF-β/SMAD3 reporter. (**A**) Cartoon depicting the TGF-β signaling cascade and activation of the reporter constructs, which are also highlighted below. MLP: major late promoter, sfGFP: superfolder GFP, dynGFP: dynamic GFP. (**B**) Images of B16F10 cells stably expressing the various TGF-β/SMAD3 reporters treated with TGF-β (1 ng/mL) for 48 h (TdTomato/GFP) or 24 h (dynGFP) or treated with SB505124 (1 mM) together with TGF-β. (**C**) Flow cytometry of B16F10 cells stably expressing the indicated reporters and treated with TGF-β (1 ng/mL) for 2, 4, 8, 24, and 48 h. SB505124 was added to TGF-β pre-treated (48 h for CAGA_12_-TdTomato/CAGA_12_-GFP, 24 h for CAGA_12_-dynGFP) B16F10 cells. The cut-off for positive cells is shown in the dashed line. (**D**) Relative percentage of positive cells calculated from (**C**). Area under the curve (AUC) for TGF-β treatment: CAGA_12_-TdTomato: 4203 ± 37.01, CAGA_12_-GFP: 3999 ± 8.40, CAGA_12_-dynGFP: 4572 ± 13.02. AUC for SB505124 treatment: CAGA_12_-TdTomato: 3974 ± 34.01, CAGA_12_-GFP: 3602 ± 40.66, CAGA_12_-dynGFP: 2001 ± 108.9. One-way ANOVA was performed on AUC values, *p* < 0.0001. (**E**,**F**) The half-life of dynGFP was measured using signal intensity on western blot (**E**) and relative mean fluorescence intensity (MFI) on flow cytometry (**F**). Cycloheximide (50 mg/mL) or SB505124 (1 mM) was added after TGF-β pre-treatment (24 h) of CAGA_12_-dynGFP B16F10 cells. Statistics for all timepoints were determined using two-way ANOVA, shown for timepoint 4 h. (**G**) The half-life of TdTomato, GFP, and dynGFP were compared by measuring relative MFI on flow cytometry. B16F10 cells were pre-treated with TGF-β (48 h for CAGA_12_-TdTomato/CAGA_12_-GFP, 24 h for CAGA_12_-dynGFP) and treated with cycloheximide (50 mg/mL). Statistics for all timepoints were determined using two-way ANOVA, shown for timepoint 8 h. All experiments were performed in biological triplicates or duplicates (**G**); error bars represent ± SEM. *** *p* ≤ 0.001, **** *p* ≤ 0.0001.

**Figure 2 cancers-14-02508-f002:**
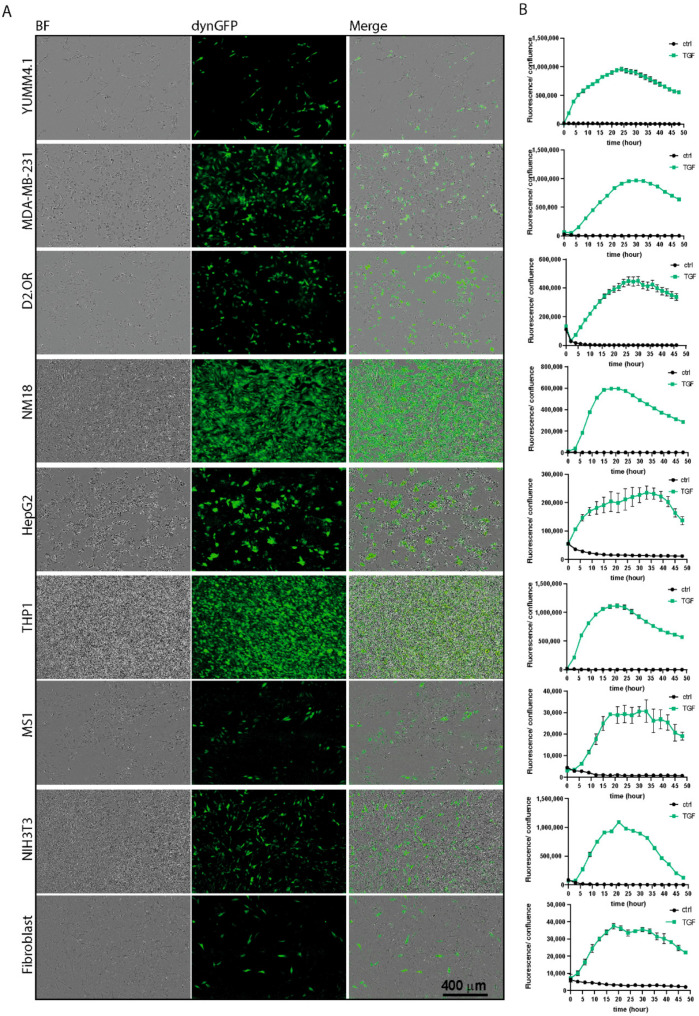
Application of TGF-β/SMAD dynGFP reporter for live cell imaging in different cell lines. (**A**) Cells expressing the pLV.CAGA_12_-dynGFP reporter (mouse normal mammary gland NMuMG derivative (NM18), human liver cancer cells (HepG2), human acute monocytic leukemia cells (THP1, suspension), mouse endothelial (MS1) and fibroblast cell lines (NIH3T3), and human primary fibroblasts) were stimulated with 2.5 ng/mL TGF-β3 or 1 ng/mL TGF-β3 (mouse melanoma cells (YUMM4.1), human and mouse breast cancer cells (MDA-MB-231 and D2.OR). Pictures were taken 24 h after stimulation on the Incucyte^®^. (**B**) dynGFP signal was measured for 48 h upon addition of control (black) or TGF-β (green) on the Incucyte^®^, *n* ≥ 3. dynGFP fluorescence signal was corrected by confluency (%). Higher background fluorescence during early timepoints on the Incucyte^®^ results in slightly elevated signals in control cells.

**Figure 3 cancers-14-02508-f003:**
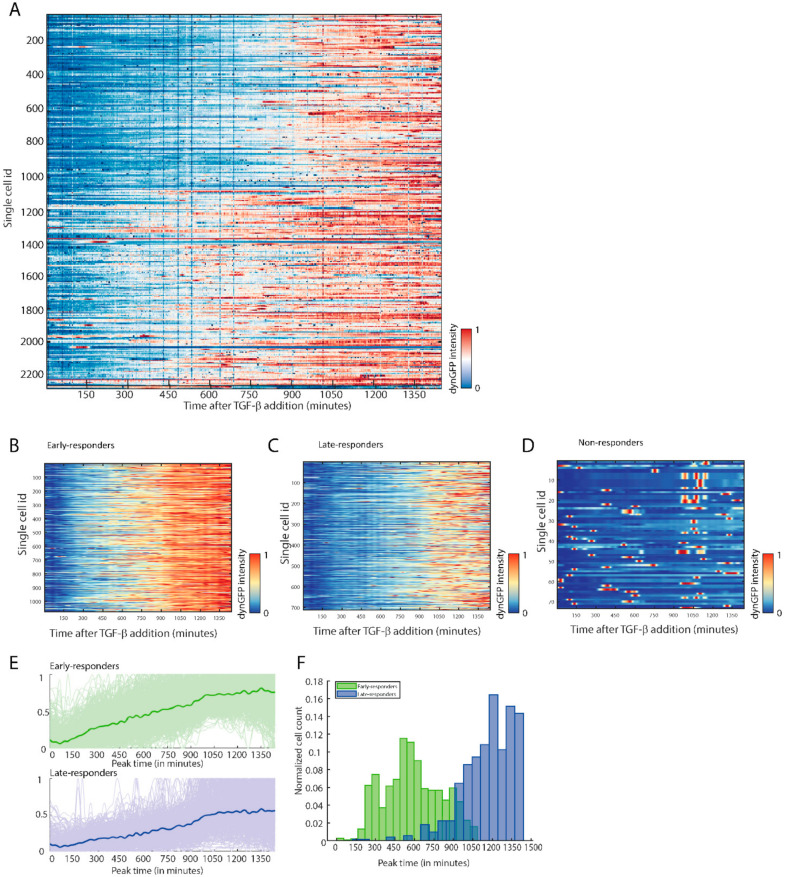
The TGF-β/SMAD3 dynGFP reporter can be used to identify heterogenous TGF-β/SMAD3 responses on a single cell level. (**A**) Heatmap showing the increase of relative dynGFP intensity between 0 (blue) and 1 (red) of live cell imaging for 24 h of individual B16F10 melanoma cells stably expressing CAGA_12_-dynGFP (Clone 7 population). Cells were stimulated with TGF-β1 (1 ng/mL) during the 24 h of imaging. (**B**–**D**) The dynamic TGF-β/SMAD3 dynGFP response in B16F10 cells can be divided into three distinct clusters: early responders (**B**), late responders (**C**), and non-responders (**D**). (**E**) Early TGF-β/SMAD3 dynGFP responders show an increase and plateau in dynGFP signal earlier after TGF-β stimulation compared to late responders. (**F**) Histogram plot reveals that the timing of the peak in dynGFP signal after TGF-β stimulation differs between early and late responders.

**Figure 4 cancers-14-02508-f004:**
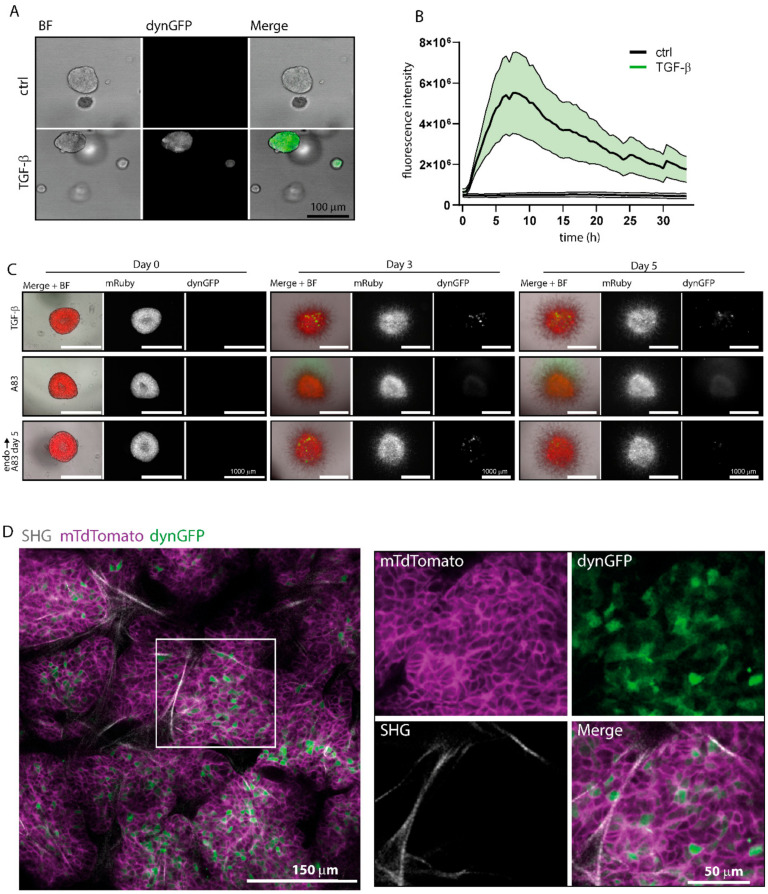
(**A**) Colorectal cancer organoids, P26T, were stimulated with TGF-β (1 ng/mL) for 7 h. Stimulated organoids show an increase in dynGFP signal. (**B**) Quantified dynGFP signal of A) ± SEM (in green) of control (*n* = 4) and TGF-β treated (*n* = 10) organoids. (**C**) Immortalized fibroblasts, RLT-PSCs, expressing H2B-mRuby and CAGA_12_-dynGFP were selected for CAGA_12_-dynGFP transduction through puromycin selection and FACS sorting. Co-cultures of RLT-PSC fibroblasts with PDAC organoids were stimulated with TGF-β1 (5 ng/mL), A83-01 (500 nM), or endogenous conditions up to 5 days. Upon TGF-β stimulation, dynGFP expression in some RLT-PSCs was observed. Upon 3 days of culturing in endogenous conditions, dynGFP expression could be found in RLT-PSCs located in the organoid core. Endogenous CAGA_12_-dynGFP could be blocked by the addition of A83-01 at day 4. Pictures are representative of *n* = 5. (**D**) PYMT organoids originating from MMTV-PyMT;R26-mTmG transgenic FVB mice were transduced with CAGA_12_-dynGFP reporter, and selected through FACS sorting. Upon transplantation of these organoids in the mammary fat pad of FVB mice, intravital imaging was performed through surgical exposure. Cell membranes were tagged with TdTomato through an N-terminal membrane tag (mTdTomato, shown in magenta); second harmonic generation (SHG) signal, showing collagen-1 fibers for structural information, is shown in grey. Green cytoplasmic dynGFP signal can be visualized. Representative image is shown.

## Data Availability

No new data were created or analyzed in this study. Data sharing is not applicable to this article.

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
