# Peer review of "Dynamic Visualization of TGF-β/SMAD3 Transcriptional Responses in Single Living Cells"

_cancers, 2022, doi:10.3390/cancers14102508_

Round 1

Reviewer 1 Report

The authors have made an impressive effort to address the points of the reviewers, removing results and/or interpretation and adding experimental data to further clarify the remaining results. Comment 19 is partially addressed as constitutive reporter was used in vitro but not in vivo but new text acknowledges limitations. There are a number of English language/style errors highlighted in the attached file. In addition: please check the colour labelling for Fig. 1F.

Author Response

Dear reviewer,

Thank you for your kind words. 

There are a number of English language/style errors highlighted in the attached file.

We have corrected the highlighted English spelling and style errors, thank you for pointing these out.

In addition: please check the colour labelling for Fig. 1F.

The color labeling in Fig 1F was indeed not correct. Our apologies, and it has been corrected in the revised manuscript.

We hope with these last adaptations on the revised manuscript, we have addressed all your concerns. We would like to thank you for your much appreciated feedback and comments on this (revised) manuscript

Reviewer 2 Report

General comments

The authors have made substantial modifications to address the issues identified in the initial reviews. This work now provides a good proof of concept for the use of this new reporter to image TGFb signaling in vitro and in vivo. However, the scope of the article would have been higher if biologically relevant information could have been drawn from the use of this tool for in vivo imaging.

specific comments:
For the single cell imaging, please give more details on the conditions used (microscope, magnification and software used for tracking).
In the material and methods section, the authors state that "the dataset could be further 431 classified into groups via Euclidean distance-based hierarchical clustering analysis". This clustering could be shown in the figure 3A for more clarity.

To address one of my previous criticisms, the authors state in the legend to figure 1 that "Statistics were calculated using one-way ANOVA". However I am not convinced that ANOVA is an appropriate method to compare time course kinetics. Furthermore, ANOVA is a parametric method, based on the assumption of a normal distribution of values, which is not suitable for scores calculated from discrete values, such as a number of positive cells (Figure 1D).

In the new figure 4B, please indicate what the green and gre areas represent : SEM?

Author Response

Dear reviewer,

Thank you for your kind words on the revised manuscript.

We have addressed your specific comments in the revised version of the manuscript.

Comment 1
For the single cell imaging, please give more details on the conditions used (microscope, magnification and software used for tracking).

We have now provided the following information in the methods section (see attached manuscript).

“The cells were simultaneously imaged using 460 nm (dynGFP; 10mW/cm2) and 532 nm (RFP-NLS10; mW/cm2) excitation for 1 day at 1 frame per 3 minutes with the 1x objective (numerical aperture = 0.5) in the Ultrawide field-of-view optical microscope[36]. “ [line 406-409]

Comment 2
In the material and methods section, the authors state that "the dataset could be further 431 classified into groups via Euclidean distance-based hierarchical clustering analysis". This clustering could be shown in the figure 3A for more clarity.

Thank you for this suggestion. We have included the figure from our reviewer response, showing the initial 10 groups, in Supplemental Figure 3C, and revised this in the manuscript attached to this reply.

Comment 3

To address one of my previous criticisms, the authors state in the legend to figure 1 that "Statistics were calculated using one-way ANOVA". However I am not convinced that ANOVA is an appropriate method to compare time course kinetics. Furthermore, ANOVA is a parametric method, based on the assumption of a normal distribution of values, which is not suitable for scores calculated from discrete values, such as a number of positive cells (Figure 1D).

We agree with the reviewer that using a ANOVA is not an optimal analysis for Figure 1D due to the reasons mentioned above by the reviewer. We have now calculated the Area under the curve (AUC) for all reporters in Figure 1D, mentioned in the figure legend, and used the AUC to calculate the statistical difference using a one-way ANOVA. This test results in a highly significant difference between the CAGA-dynGFP reporter and the CAGA-TdTomato/GFP reporter, in both the TGF-b treatment and SB505124 treatment panel. These changes are highlighted in the attached manuscript in red. We believe that this is a better method to determine the significance in Figure 1D, and we hope the reviewer agrees.

Comment 4

In the new figure 4B, please indicate what the green and gre areas represent : SEM?

Thank you for notifying us that this was missing in the Figure legends. We have now included in the figure legend that the green area represent the SEM.

We hope that with these additional revisions, we have satisfied this reviewer comments on the (revised) manuscript.

Reviewer 3 Report

The authors responded sincerely to a lot of points raised by reviewers. The revised manuscript can be accepted. 

Author Response

Dear reviewer,

Thank you for your kind words on the revised manuscript. We are glad to have addressed all your concerns about our revised manuscript.

This manuscript is a resubmission of an earlier submission. The following is a list of the peer review reports and author responses from that submission.

Round 1

Reviewer 1 Report

This manuscript describes the development and use of a novel fluorescent reporter construct driven by the TGF-beta/SMAD3-dependent CAGA12 promoter found in other reporters. The construct is engineered to express super-folding GFP with a PEST and is more rapidly detected upon TGF-beta treatment and its detection more rapidly lost upon treatment with a TGFBR inhibitor, as compared to GFP and TdTomato constructs. The authors test this construct in multiple cell lines in 2D and 3D, patient-derived organoids, organoid co-cultures and using intravital imaging of MMTV tumors in mice. The results of live cell imaging experiments using single cells leads the authors to propose that the reporter detects cell cycle-related differences in the response to TGF-beta.

In general, interesting observations have been made with this reporter but these are diminished by some lack of detail and controls, particularly with respect to the organoid and in vivo experiments and the link to the cell cycle, which all require additional experiments/controls to support the conclusions drawn.

Specific comments

Methods 1. The authors need to rule out the possibility that heterogeneity is independent of transduction efficiency, for example using dynGFP driven by a constitutive promoter, e.g., CMV. 2. They should clarify when the cells/organoids were selected with puromycin after transduction and when they were sorted by FACS. 3. They should clarify when and where they used A83-01, a TGF-beta receptor inhibitor, in organoid cultures and, if present, explain how this affected the experiments involving TGF-beta stimulation/endogenous activity monitoring. 4. Single cell imaging shows 100,000 cells were used but later 2,200 and 9,000 are shown in the figures. Are the authors able to track only <10% of the population? 5. The cells appear to be quite confluent, how efficient was cell segmentation/masking? Some examples need to be presented.

Results 1. Fig 1C/D shows 48 h after 1 ng/ml stimulation with TGF-beta, dynGFP signal remains high/stable, however in Suppl. 1D for the same cells (B16F10), the signal at 48 h using 2.5 ng/ml is low/null. This was using a different method but how can these results be reconciled? 2. Fig. 1D vs 1E - labelling confusing - SB505124 treatment shows no reduction at 8 h in D but reduction of 30% in E (?) 3. Line 467 - removal of TGF-beta mentioned but not in the text and not in the figures (?) 4. Suppl. Fig. 1D – was the GFP signal homogeneous among the population or heterogeneous as is later indicated for other models? 5. Line 494 - should this refer to fig. 1D? 6. lines 517-518 - protein “half-life” of the dynGFP compared to normal GFP as in fluorescence intensity? This requires a CHX experiment directly comparing standard GFP and TdTomato constructs. 7. Fig. 2 – the graphs are a little confusing. 8. A) dynGFP 0,661 correlation compared to what as a control? Is there repeated data presented in A and B? 9. Please review the use of Serpine1 versus PAI1 nomenclature, Suppl. Fig. 2D indicates PAI1 RNA 10. line 555-563 – the delay calculations described should be shown in the supplement with overlapping shifted graphs. Indicate the significance of the correlation in “2 hour delay (Pearson correlation 0.9222)” line 561, as done later (569-570). 11. Line 570 – Refers to Suppl. Fig. 2C/D? Because from Suppl. Fig 2 only B is cited in the text. Clarify in the text figure of Suppl. Fig. 2C/D if data corresponds to stabilized CAGA12-GFP reporter, as this is confusing. 12. Line 579 – what do the authors mean by efficiency? 13. Fig. 3A – line 594 - why is no endogenous signaling observed. Is it because the reporter is not sensitive enough to detect it, or because it is being inhibited by media components? 14. Fig. 3B – it is difficult to see these images and distinguish PDX from fibroblasts. In order to establish this as endogenous signal, a recovery experiment is needed, for example after use of the TGF-beta inhibitor SB505124 or by keeping A83-01 in the media, if it was removed. 15. Fig. 3C – a magenta only image should be included. Is heterogeneity of the GFP signal independent of transduction efficiency? Do the PyMT organoids contain infiltrating mouse cells that affect the signal? A control promoter-driven dynGFP reporter is required. 16. Suppl. Fig. 3 – The % GFP signal over confluence should be shown, as done in Suppl. Fig. 1D, to give an indication of endogenous TGF-beta signaling activity in the cell lines used. 17. Fig. 4 – are the results shown in A, C-E from the same experiment? How many times has the experiment been carried out in this line? It would be helpful to also show the data in A clustered, and in that way not lose visualization the % of population corresponding to each cluster. Is there an additional cluster with cells not associated with any of these behaviors? 18. Line 664 – “independent clone (Figure S4A, B) with similar results.” Are then these percentages of early and late responders conserved in the other clone (Suppl. Fig. 4)? 19. Fig. 5A – y-Axis label missing 20. Suppl. Fig. 5F/G text missing. 21. The authors should discuss the implication of the previously calculated delay in the GFP signal (0-4 h) after the TGF-beta stimulation and the delay in the signal decay in the context of the cell cycle analysis. Do the claims for the different cell cycle stages still hold if the TGF-beta reporter only shows delayed activation and inactivation? 22. Additional experiments using synchronized cells could help support the cell cycle specificity of the signal. Importantly, the authors need to rule out that the cell cycle effects are not related to cell mechanisms of dynGFP synthesis and/or degradation. This requires use of a construct expressing dynGFP driven by an unrelated (not TGF-beta-regulated) promoter. Further support could be obtained by looking for cell cycle regulated expression of SERPINE1.

Discussion 1. 762 – Half life is improved, meaning it is reduced? 2. 776 – Serpine RNA levels follow 2 h later or prior? 3. It would be useful to discuss autocrine vs paracrine TGF-beta signaling and discuss the limitations of this novel construct in more detail.

General points 1. Limited timepoints make it difficult to establish the full potential of dynGFP construct, as already acknowledged by the authors, a middle timepoint between 8 and 24 h is required. With the availability of the Incucyte, this could be readily done. 2. The conclusion line 572 should be clarified “in B16F10 cells¨ 3. Using a clonal cell population for the TGF-beta response makes it less relevant for the in vivo situation/heterogeneity point. In fact, a different mechanism is proposed in the discussion for the heterogeneity found in the tumors, linking it to EMT or metastasis, more than to cell cycle (lines 807-808). 4. An experiment like the one shown in Fig. 5G, for analysis of TGF-beta target gene expression in GFP high and GFP low FACS-sorted populations would be helpful. This would be most interesting in the organoid cultures and in the tumors where a marked heterogeneity is shown.

Minor points/suggestions Fig 1 / Suppl Fig 1 – SB505124 is always added in combination with TGF-beta, so TGFB+SB in figure legends might be less confusing. Fig 1E – the control line is difficult to see. Suppl. Fig 1E – should go into the main figure as it helps establish one of the main conclusions. Typos (line): 30 – missing point; 21, 50 – proto-typical no hyphen; 176 – domein; 230 – Magrigel; 241 – stablely; 317 – leammli; 401 – were>was; 416 – ms > min; 476 – transcriptional; 527 – investigated; 597 – vivo > in vivo; 806 – could suggesting. There should be spaces between all numbers and units. Clarify some abbreviations: 94 – SBE, SMAD binding element; Fig. 1 – FI, Fluorescence intensity; MFI – Mean Fluorescence Intensity. All in vitro and in vivo should be italicized. Fig1A it would be helpful to show the TGFBR complex traversing a membrane.

Reviewer 2 Report

In this manuscript Marvin et al successfully developed a dynamic fluorescent reporter of TGF-beta signaling which can be used both in vitro and in vivo. This reviewer only has minor questions.

  1. Fig S1C is not referred in the main text.
  2. Page 11 line 504, the authors concluded that downstream of block by SB505124, p-Smad still induces target gene expression. However, in Fig 1F, pSmad2 is no longer detected at 2h, early after addition of SB. Would it be better explained by the delay of dynGFP reporter as shown in Fig.2?
  3. Regarding the discrepancy between fig 1D and E: at 8h after addition of SB, dynGFP positive cells were still about 100%, however, total fluorescence in 1E already went down to 20-30%.
  4. This reviewer is concerned about the selection of top 20% GFP positive cell population after 48h TGF-b addition for establishing B16F10 cells with dynGFP in page 5 line 217. This may cause selection bias so that strongly TGF-beta responsive cells at later phase after stimulation are enriched and affect the results in Fig. 4 and 5, for example.
  5. Fig 1A. Is there linker sequence between sfGFP and PEST? If so, the information should be provided in the method. If not, sfGFP needs to be directly followed by PEST in the scheme.
  6. The authors should indicate which isoform of the TGF-b was used for each figure.
  7. Fig 3C. As a positive control to organoid culture, is the endogenous TGF-beta signal detected by dynGFP in the case of cancer cell line? This point was not described in the previous figures.
  8. Fig 4E. Why there are no responders? DynGFP positive B16F10cells were selected after 48h TGF-beta stimulation in the method. Do the no responders reside in any of the cell cycle points? No responders were not evaluated in Fig. 5.
  9. Regarding the relationship between cell cycle status and dynGFP response, have the authors evaluated by synchronizing the cell cycle?
  10. The data suggested that cells in G2-M phase show faster GFP signal. This reviewer wonders whether cells in M phase are able to respond to the signal. Have the author checked the presence of GFP signal stimulated in mitotic phase?

Reviewer 3 Report

Major points:

-1- No relevant biological informations can be drawn from in vitro ad in vivo 3D models in figure 3. Eg: what is the phenotype of responsive versus non responsive organoids in figure 3A ? What is the effect of responsive or non-responsive PSCs on neighboring PDAC organoids ? What is the dynamic of TGFb signaling in the tissue micro-environment (Figure 3C) and the link with surrounding tissue features (SHG signal) ?
-1- The authors strongly suggest that the heterogeneity of fast versus late responsive cells can be driven by cell cycle phases, but there is no real demonstration of this hypothesis.

Lane 218 CAGA-reporter cell lines are sorted (FACS) after stimulation with TGFb3. What is this choice of using TGFb3 based on ?
Treatments were performed using either TGFb or TGFb3 without expliciting the reasons of using such or such TGFb isoform.

Reference 35 is not yet available (in press). mTGMM cell tracking algorithm should therefore be provided in more details.

lane 416: Early responders are said "going up around 300 ms" while time scales in figure 4 are in min, not in ms.

In control experiments (Figure S1) the authors choose to monitor SMAD2 ohosphorylation, while, as mentionned in the introduction, SERPINE1 CAGA boxes bind SMAD3/4 heterodimer. It would be therefore more relevant to monitor SMAD3 phosphorylation. The choice of showig rather SMAD2 phosphorylation should be explicited.

In figure S1D, signal increases are commented but not spontaneous signal decreases (without adding inhibitor) observed after various times of TGFb treatment.

Figure 1D: statistic test used to calculate the p-values from cell counts should be indicated.

lane 557: replace "Serpine" by "Serpine1".

lanes 505-506 and 518-520. Suggests that the dynamic response is limitated by mRNA stability rather than protein turnover. When discussing a way of improving dynamics by tuning mRNA half life, the author should consider adding decay elements such as AREs in the 3'UTR.
In the same line, can the different dynamics between Smad7, Ctgf and Serpine1 (figure 2) arise from different mRNA halflives ? This point might be discussed in more details.

Figure S2 A,B: differences between S2A (dynGFP) and S2B (GFP) are not commented, neither in the figure legend, nor in the text. This is an important point because dynamics of GFP versus dynGFP decreases look very similar when comparing S2A and S2B, whereas dynGFP decrease seems to be faster than GFP when comparing FACS data (compare figure 2C and S2E/F). The authors should comment this discrepancy.

Refering Serpine1 mRNA as either "Serpine1 RNA" (Figure 2B/C) or "PAI1 RNA" (figure S2) is confusing, so is the legend "PAI1" of figure S2D,F which deals with Serpine1 mRNAs, not PAI protein.

The paragraph dealing with expression delays between (dyn)GFP proteins and mRNAs (lanes 555 to 570) is not easy to understand since Pearson's correlation tests (either coefficients or p-values) compare data from two distinct figures (2 and S2) and statistics discussed in the text body are not indicated on these figures. This part might be more understandable if figures 2 and S2 were put together, with the results of Pearson's comparisons.

Figure 3A: the result would be more convincing if a polulation analysis (number of responding versus non-responding spheroids) were provided.

Figure 3B: PDAC organoids semm to express mRuby as well. If true, this should be mentionned in the legend.

Figure 3C: there is no comment about the SHG signal in xenografts, neither in the legend nor in the text. What is the purpose of SHG monitoring ? What conclusions can be drawn from this signal ?
This experiment laks a time course before and after transplantation to assess the dynamic contribution of tumor microenvironment to dynGFP signal. In fact, there is no real control to accurately measure signal/noise ratios, and to assess the dependancy of dynEGF signal with respect to the host tissue TGFb signal.

Figure 4E: the "non-responding" cells exhibit some transient signals over time. Are these pulses of dynGFP signals artefactual or do they correspond to some biological fact ? This question should be addressed by the authors.

Given the sharp difference between early (~ 1000 cells) and late (~ 700 cells) responsive cells, the overall response of a cell population over time would be expectedly biphasic, which is not observed in figure S1D. This point should be discussed.

Figure S4: The time duration of TGFb treatment is not indicated.

lane 697: the sentence "...separating the early and late responder clusters...) suggests that these two populations will be compared in figure 5. In fact, only cells treated for 2 hours are considered in figure 5 which compares responsive and not-responsive cells at this time. This sentence should therefore be changed for more clarity. This observation therefore does not support the hypothesis proposed lanes 708-711, of a causal link between cell cycle and responsiveness to TGFb.

Discussion about the delay between dynGFP signal and Serpine1 expression is not useful to my point of view, unless a clear message would arise by fusing data from figures 2 and S2.

The heterogeneity of responses in cultured cells (early vs late responding cells and non-responsive cells) is one of the most interesting aspects of this study and underlying mechanisms should be discussed more deeply: do the authors propose that cell cyle is the only or major driver of this heterogeneity ?

Reviewer 4 Report

In this study, the authors developed a dynamic TGF-beta/SMAD3 transcriptional reporter of GFP that can quickly reflect the signal transduction at a single living cell level. It can be used not only for in vitro, but for in vivo transplantation model.

The first question is, why did the authors choose this journal? Probably this reporter can be used not only for cancer cells but wide application. Journals related to molecular cell biology etc. might be more suitable.

TGF-beta/SMAD signaling is well known to inhibit epithelial cell proliferation through G-1 cell cycle arrest. In this study, the authors analyzed cell cycle, but did not observe G-1 arrest. Cancer cells might escape from the TGF-beta-induced cell cycle arrest, but, how was the cell proliferation in these cell lines after the treatment with TGF-beta? I think that this reporter might be more useful for non-cancer epithelial cells and to see the correlation of TGF-beta signaling with cell cycle arrest and cell proliferation.

Minor points

In Figure 3, organoids were stimulated with TGF-beta for 7 days, which might be relatively long time incubation. Was the medium as well as TGF-beta refreshed during the incubation?

In Figure S5, legends of (F) and (G) are lost. In Figure 4, style of legends of (F) and (G) is different from (A) to (E), which is a bit difficult to read.